# Psychometric Study of "Questionnaire of Barriers Perceived" (QBP) in Higher Education

**María Dolores Hidalgo-Ariza, Eva Francisca Hinojosa-Pareja** 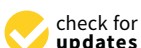 **and Juan Manuel Muñoz-González ***

Department of Education, Faculty of Education Sciences, University of Córdoba, 14004 Córdoba, Spain;
lola.hidalgo@uco.es (M.D.H.-A.); ehinojosa@uco.es (E.F.H.-P.)
* Correspondence: juan.manuel@uco.es

**Abstract:** This article presents the process of adaptation and validation, and the resulting psychometric properties, of the "Questionnaire of Barriers Perceived" (QBP). The scale identifies whether a student's perceptions and expectations are mediated by stereotypes or roles associated with gender through the study of their professional aspirations, fear of negative judgement, and perceptions/awareness of gender roles of men and women. Two descriptive studies were conducted via a cross-sectional poll. The questionnaire was administered first to 240 students and then to a total of 1044 student from all the degrees studied at the Faculty of Education at the university at which the study took place. The data were subjected to item content analysis, descriptive analysis, analysis of internal consistency, study of the relationship between variables, correlational analysis, and an exploratory and confirmatory factorial analysis. The results showed that the scale had a high goodness-of-fit index, as well as validity and reliability. The dimensions that the model comprised were found to be interrelated and coherent with the theoretical structure considered in the initial version of the instrument. The resulting questionnaire presented sufficient validity and reliability to be used in other contexts and studies of the same nature.

**Keywords:** instrument; validation; stereotype; higher education

## 1. Introduction

Recent studies have confirmed the persistence of barriers related with gender that make difficult the educational and professional development of women, both in general and specifically in the area of education (Donoso et al. 2011; Porto et al. 2012; Martínez Labrín and Urrutia 2013; Castillo and Montes 2014; Hidalgo-Ariza and Muñoz-González 2017; Barone et al. 2019; Mauerer and Schmidt 2019). As a result, at the theoretical level (Contreras Hernández and Trujillo Cristoffanini 2014), as well as the regulatory level (Ley 2007, de 26 de noviembre, de Medidas de Prevención y Protección Integral contra la Violencia de Género (Law 13/2007, from November 26, Measures for the Prevention and Comprehensive Protection against Gender-based Violence); Strategic engagement for gender equality 2016–2019, European Commission 2016), there is an emphasis on the need to visualize and question the barriers that are implicitly and explicitly found in social, academic, and professional sectors.

Gender inequality and the efforts made towards its eradication are not new or recent. Starting with the feminist movements, there has been an emphasis on the fight for equality of opportunities between women and men, an emphasis that has been conveyed by some government agencies that have echoed this need by promoting initiatives in favor of equal rights and the elimination of inequalities. Examples of this are the eight Millennium Development Goals (MDG) established by United Nations to be met in a period of 15 years from 2000 to 2015, among which was the goal "to promote gender equality and empower women". Once the objectives were reviewed after this period had passed, the conclusions were discussed in a report, in which it was established that despite reaching a level of

significant success, gender inequality persisted in many regions and spheres of life. Thus, a new agenda was set in which new Sustainable Development Goals (SDG) were established to be met in 2030, among which was the goal to "achieve gender equality and empower all women and girls" (UNESCO Incheon Declaration 2015).

What elements make difficult or obstruct effective equality in the professional opportunities of women and men? Despite the complexity of this question, which is due to its historical and contextual characteristics linked with cultural, social, economic, political, identity-related, psychological, etc. elements, different research studies have tried to identify factors or barriers that could have an influence on the professional and academic choices and trajectories of men and women. These barriers are associated, among others, with stereotypes and roles attributed to genders, which are acquired and consolidated throughout life through different socialization agents. The works by Miller et al. (2015) and Colás Bravo and Villaciervos (2007) showed this, evidencing gender stereotypes possessed by students from different educational stages. These stereotypes refer to areas such as the body, social behavior, competencies and abilities, emotions, affective expression, and social responsibility (Colás Bravo and Villaciervos 2007) or responsibility in the professional sector (Miller et al. 2015). With respect to these areas, the people who participated in both research studies assigned stereotypical and differential characteristics to women and men. Particularly, in the academic and professional sector, it was shown how boys and girls possessed the belief that girls had a greater aptitude for humanities and social sciences degrees and boys for technical subjects and science.

These stereotypes, together with other conditioning factors that were also cross-examined by gender such as the perception of aptitudes, interests and motivations, or pressure from the family and social sector, determined the academic and professional orientation of students, and consequently their choice of a professional occupation (Meece 2006; Barberá Heredia et al. 2011; Olivares García and Olivares García 2013; Pérez-Carbonell and Ramos Santana 2015; Mau and Li 2018).

Numerous studies (Muñoz et al. 2011; Arias and Bascón 2012; García Gómez 2013; Yazilitas et al. 2013; Larrieta et al. 2015; Guil Bozal 2016; Eslen-Ziya and Yildirim 2022) have determined that women suffer a double segregation in the academic and professional sectors: horizontal and vertical. Per horizontal segregation, some areas of knowledge are too feminized, such as education, health sciences, and humanities (related to caretaking and social services), while others are too masculinized, such as engineering, sciences, and math (more technical and mechanical areas). Martínez (2013) pointed out, in this respect, that female students opted to study teaching, nursing, or psychology while male students leaned towards computer science, sports science, or engineering. These different trajectories showed not only different interests as a function of gender, but a social and professional differentiation for women and men, as well as unequal options related to their social and economic value in terms of prestige or even pay.

On the other hand, there also exists vertical segregation, as the number of women occupying positions of responsibility are scarce despite being equally qualified to occupy them. In fact, paradoxically, different studies have indicated that the rate of school failure and dropout is higher for men, while women have a higher level of training and achieve better academic results (O'Neil and Bilimoria 2005; Hopkins and O'Neil 2007; Barberá Heredia et al. 2011; Selva Olid 2012; Vera Gil and Guil Bozal 2014; Larrieta et al. 2015). These research studies have argued that this difference is due to women's greater resilience, meaning the ability they possess to overcome the difficulties they may find. Also, women have shown greater levels of motivation and constancy, and have been shown to be tougher in the judgements they make of themselves (Torres-Guijarro and Bengoechea 2016), which makes them more disciplined. As a result, they achieve higher academic grades. In spite of this, it has been evidenced that they have greater difficulties in their incorporation in the professional world and their advancement within it.

This vertical segregation is associated with the existence of a "glass ceiling" that Guil Bozal (2016) defined as the set of barriers women find for accessing higher positions,

such as the difficulty in reconciling work with maternity, the demands of family, or the social expectations that define the proper and accepted behaviors for women and men. This "glass ceiling" brings with it, in many cases, renouncing of options for higher, more prestigious positions that are better paid, and ultimately not advancing at work. In this vein, authors such as Cubillas Rodríguez et al. (2016) have highlighted that although the change in woman's role related to their participation in the area of work has been accepted, in their private lives, women are still the ones who take on the family responsibilities, household tasks, and raising of children. Also, traditionally, as shown by Blackmore and Sachs (2007), entrepreneurship has been associated with men, conferring men with positive value. However, it had negative connotations when the leadership was associated with women, as it was thought that men had more qualities conducive to leadership and that leadership was not the correct environment for women.

Contreras Torres et al. (2012) added to the barriers mentioned biased systems of evaluation and promotion, a lack of reference models of women in positions of leadership, and scarce informal social networks for women. Ultimately, the inequalities women face are due to unequal structures and organizational processes, as well as a lack of measures in favor of equality of opportunities.

There are different instruments that have been used to measure factors related with gender barriers in academic and professional sectors in the national and international context. Many of these instruments analyze the presence of stereotypes in the attitudes, positioning, or predisposition of the participants.

Bergman (2003), for example, conducted a study on health, psychological stress, and satisfaction at work of women through a questionnaire that used a scale with four options for response. Validation was performed through an exploratory factorial analysis (EFA) using a principal component analysis with varimax rotation. The results consisted of four factors: barriers perceived by women, obstacles in their personal experience, sexual harassment, and difficulties in organizational structure. The reliability of the instrument was measured with Cronbach's alpha coefficient.

De Sola et al. (2003) conducted an analysis of attitudes towards gender equality using a questionnaire with a Likert-type response scale composed of seven options. Validation was conducted with a principal component analysis, through which a total of six factors were obtained: sexual orientation, religious symbolism, values and couple's structure, private life, public sector, and sexuality and personal freedom. The reliability of the instrument was measured through a study of its internal consistency using Cronbach's alpha coefficient.

Baber and Tucker (2006) studied attitudes towards gender with a social role questionnaire (SRQ) that used Likert-type answers with four response options. Validation of the instrument was conducted through a confirmatory factorial analysis, taking into account convergent and discriminant validities, through which a total of three factors were obtained: a general view of social roles, the awareness of children about these roles, and lastly, the importance of gender. The questionnaire's reliability was measured through Cronbach's alpha coefficient.

García Pérez et al. (2010) measured the degree of the student's predisposition towards equality and coeducation with the use of a scale, designing and validating, to this end, the SDG/s (School Doing Gender/students) questionnaire. This scale was designed ad hoc, included a Likert-type response system composed of five options, and comprised three main dimensions: sociocultural, relational, and personal. Validation was conducted with an exploratory factorial analysis (EFA) through principal component analysis; reliability was measured with Cronbach's alpha coefficient. Also, in a similar study conducted by Rebollo-Catalán et al. (2011), the attitudes of the teachers towards coeducation were examined using the same method as that in the previous study described, but including qualitative group discussions.

On their part, Bergman et al. (2014) developed a study about men's beliefs on gender inequalities in the work sector. More specifically, the study was focused on the validation of a questionnaire called the "Men's Polarized Gender Thinking Questionnaire (MPGQ)"

through the use of a confirmatory factorial analysis (CFA) along with a maximum likelihood (ML) analysis, which resulted in six factors: different points of view on success, stereotypical gender roles, benevolent sexism, awareness of the importance of gender, knowledge about a male norm system, and strategies for gender equality. The reliability of the instrument was measured through Cronbach's alpha coefficient.

Mosteiro García and Porto Castro (2017) analyzed the gender stereotypes possessed by vocational training students, designing to this end an ad hoc scale (Likert type, with five response options) that was validated with a group of experts. The reliability of the instrument was again verified through Cronbach's alpha coefficient.

Lastly, Lajo et al. (2008) and later Donoso et al. (2011) investigated the gender barriers in the professional development of university students. For this, the authors created a questionnaire that brought together five dimensions: leadership aspirations, relevance of the role of work, fear of negative judgement, attitude towards social networks, and awareness of the role of gender. Validation of the instrument was conducted through expert judgement, with the reliability measured with Cronbach's alpha coefficient.

Among the questionnaires and scales described, the latter was the only one located in a higher education context that was specifically concerned about barriers associated not only with attitudes, but with other aspects such as aspirations or fears, which are determining issues when choosing an academic or professional career. However, in its validation process, neither its internal structure nor its coherence with the theoretical assumptions considered in its construction were analyzed.

This article contributes in this regard, presenting the process for its adaptation and validation for its use in a new research context and contributing with a double study (pilot and definitive use of the instrument) to the initial analysis of its characteristics, which resulted in the examining of its internal consistency, the improving its content, and the analysis of its internal structures in an exploratory and confirmatory manner.

This article is part of a broader research project in which a study was conducted on the gender barriers possessed by students in all the degrees from the area of education in higher education. In this sense, the study identified whether students' perceptions and future expectations were mediated by stereotypes or roles associated with gender through the study of students' professional aspirations, fear of negative judgement, and perceptions/awareness of gender roles of men and women. Likewise, an analysis was conducted on the question of differences in these perceptions as a function of different sociodemographic, academic, and family life variables and the different predictive models of the dimensions studied.

## 2. Materials and Methods

The present article is focused on the process of validation and analysis of the psychometric characteristics of the instrument "Questionnaire of Barriers Perceived" (QBP) (Appendix A). For this, as recommended by Alaminos and Castejón (2006), two studies were conducted. One was designed as a pilot study, and the other was a confirmatory study with the final sample for the research. The methodology chosen for both was a quantitative cross-sectional survey, because of the numerical and reliable nature of the data gathered, and for following a deductive and structured method of research.

### 2.1. Instrument

The instrument chosen for the research was the Questionnaire of Barriers Perceived (QBP) (Lajo et al. 2008), initially designed by the DONA/GREDI group within the research framework financed by the l'Institut Català de les Dones "Gender Barriers in the development of the female university student: the invisible filter".

In its final version, it was an anonymous polytopic questionnaire administered online with closed-ended questions that used a Likert-type response scale with five options, which ranged from total disagreement (1) to total agreement (5) in the "Professional aspirations" dimension and "Fear of negative judgement" and from none (1) to many (5) in

the dimensions "Perception of the gender role of women" and "Perception of the gender role of men". It included the independent academic variables (degree, year, current living arrangement) and family-related variables (level of education, work status of the father and the mother). It comprised a set of 63 items presented as statements and was structured in four dimensions:

- Professional aspirations: This dimension referred to the importance given by students to the professional sector as a key issue in their life and students' predisposition towards leadership, meaning their aspirations to be promoted within an organization. It encompassed 16 items that referred to commitment with a professional project, the meaning granted to work success, personal involvement, time invested to achieve professional objectives, and intent to ascend in a professional career and be recognized as a leader in the work sphere.
- Fear of negative judgement: This nine-item dimension referred to rejection in the valorations by other people and worry, negative expectations, or anguish about negative judgements by others.
- Perception of gender roles (double dimension—of women and of men): this double dimension included different characteristics of people in the professional sector related to expectations, stereotypes, and socially assigned roles for the female and male genders. It was divided into two scales differentiating between those characteristics that the students polled attributed to women and those they assigned to men. In both dimensions, the same items appeared (19 in each case), which referred to issues such as ability to plan, self-control, the assumption of responsibilities, emotional stability, interpersonal communication, facing of risks, reconciliation, the desire to succeed, and self-esteem.

### 2.2. Sample

The study context was made up of all the education degrees of the University of XXX, that is, the degrees in early childhood education, primary education, and social education, as well as the Master's degrees in inclusive education and the teaching staff of compulsory secondary education and baccalaureate, professional training, and language teaching during the 2016–2017 academic year. Therefore, it should be noted that these degrees are feminized, that is, there is a greater predominance of women over men, an aspect that is reflected in the description of the sample and that, to a certain extent, constitutes a limitation of the present study.

In both studies, the participants were selected by a purposive sampling (Etikan et al. 2016), as the instrument was administered to students who voluntarily, with consent, wanted to answer the questionnaire in each of the academic years and groups that were the object of study.

#### 2.2.1. Study 1

The first study sample was composed by a set of 240 students, of which 68.5% (165) were enrolled in the children's education degree, 7.5% (18) in primary education, and 24.1% (58) in social education. Most were women (88%, $n = 212$) and were enrolled in their second year (61.8%, $n = 149$). As for their ages, the largest group, 107 people, were 18 to 20 years of age (44.4%), followed by students aged 21 to 23 (39.4%, $n = 95$), with the last group being older than 24 (16.2%, $n = 39$).

#### 2.2.2. Study 2

A total of 1044 students participated, which corresponded to 45.99% of the population. Of this set, 345 belonged to the children's education degree (33%), 542 to primary education (51.9%), 93 to social education (8.9%), 28 to the Master's in inclusive education (2.7%), and 36 to the Master's in teaching in secondary education and baccalaureate, vocational training, and teaching of Languages (3.4%). The sample was mostly composed by women (80.7%, $n = 842$) aged from 18 to older than 26 with the highest percentage found in the

18–20 age range (41.6%, *n* = 434). As for their academic year, in the case of the Bachelor's degrees, they were closely distributed, with similar percentages among the four academic years considered: 22.0% were enrolled in the first, 28.7% in the second, 24.2% in the third, and 18.9% in the fourth year.

### 2.3. Data Analysis

2.3.1. Study 1

The first study consisted of a pilot study with the instrument in order to adapt and conceptualize it to the population studied. This pilot study offered the possibilities of detecting difficulties in the understanding of some items and of collecting of information that could allow the contrasting of the technical characteristics of the instrument in previous studies (Lajo et al. 2008) with those in the present study in order to perform new analyses to improve the instrument's validity.

To apply the questionnaire, an online version was created that facilitated filling it out. The procedure was followed up on by the researchers through the tool created online as well as in person so that they could detect difficulties in understanding it and clarify doubts during the filling-out process.

Once the information was collected, the content of the items was analyzed, and the drafting of some items was modified; also, the structure of the instrument was studied through an exploratory factorial analysis (EFA) using a polychoric correlation for its execution along with a process of "nonweighted least squares" extraction of common factors with a weighted oblimin rotation, as the data were non-normal ordinal data with a relatively small sample as compared to the number of items that had to be analyzed. The instrument's internal consistency was also analyzed with Cronbach's alpha. For this, the SPSS 23 and Factor Analysis (10.5.02) software programs were used.

2.3.2. Study 2

Once the characteristics of the instrument evidenced in study 1 were analyzed, and once it was rewritten according to the results obtained, a second study was conducted with the final research sample. The procedure for the gathering of information followed the same guidelines as in study 1.

In this case, the information obtained was analyzed statistically with basic descriptive analysis (central tendency and dispersion measurements), analysis of internal consistency (Alpha of Cronbach), study of the relationship between variables through comparison of means as a function of the variable "sex" (through the analysis of the variance with the Mann–Whitney U test), correlational analysis (Spearman's correlation coefficient), and confirmatory factorial analysis (CFA, with maximum likelihood as a method of estimation). The statistical programs used were SPSS 23 and AMOS.

### 3. Results

#### 3.1. Study 1

3.1.1. Study of the Structure of the Instrument: Exploratory Factorial Analysis (EFA)

Before the application of the test, the criteria for its viability were verified (determinant of the correlation matrix of 0.00; KMO = 0.74; Bartlett's test of sphericity with a significance of 0.00; RMSR = 0.05). Once the criteria were verified, the EFA was applied, adjusting the factors to be extracted to five.

The analysis showed that the factors extracted explained 38.08% of the variance. The commonalities oscillated between 0.03 in item 19 and 0.580 in item 59, with 33 being under 0.3 (Costello and Osborne 2005). These results, together with the difficulty in the theoretical interpretation of the rotated factors matrix, urged the review of those items that had a smaller commonality. Of these 33 items, 29 were eliminated, as they theoretically seemed to contribute expendable data to the study.

A new analysis was conducted once the variables were eliminated. The method of extraction and rotation was maintained, this time adjusting the factors to four, as 10 of

the 29 items eliminated comprised a complete dimension. The factors extracted explained a variance percentage of 47.40%. Observing the matrix of rotated factors and the weight of each item per factor (Table 1), an important relationship could be observed with the different dimensions considered in the study in practically the entire set of items with loads higher than 0.3. This was the case except for two variables (v42 and v61), which were kept after considering that their load was close to this value and that their content was relevant to the research.

**Table 1.** Matrix of rotated factors.

| Variable | F 1 | F 2 | F 3 | F 4 |
|---|---|---|---|---|
| V 1 | | | 0.66 | |
| V 2 | | | 0.35 | |
| V 3 | | | 0.36 | |
| V 4 | | | 0.52 | |
| V 5 | | | 0.46 | |
| V 6 | | | 0.54 | |
| V 7 | | | 0.65 | |
| V 8 | | | 0.51 | |
| V 9 | | | 0.59 | |
| V 10 | | | 0.60 | |
| V 11 | | | 0.57 | |
| V 12 | | | 0.66 | |
| V 13 | | | 0.51 | |
| V 14 | | | 0.45 | |
| V 15 | | | 0.44 | |
| V 16 | | | 0.58 | |
| V 17 | | 0.71 | | |
| V 18 | | −0.45 | | |
| V 19 | | 0.62 | | |
| V 20 | | −0.46 | | |
| V 21 | | 0.62 | | |
| V 22 | | 0.72 | | |
| V 23 | | −0.52 | | |
| V 24 | | 0.67 | | |
| V 25 | | 0.52 | | |
| V 26 | | | | 0.50 |
| V 27 | | | | 0.46 |
| V 28 | | | | 0.49 |
| V 29 | | | | 0.55 |
| V 30 | | | | 0.64 |
| V 31 | | | | 0.50 |
| V 32 | | | | 0.52 |
| V 33 | | | | 0.52 |
| V 34 | | | | 0.57 |
| V 35 | | | | 0.35 |
| V 36 | | | | 0.52 |
| V 37 | | | | 0.49 |
| V 38 | | | | 0.34 |
| V 39 | | | | 0.77 |
| V 40 | | | | 0.70 |
| V 41 | | | | 0.67 |
| V 42 | | | | 0.28 |
| V 43 | | | | 0.40 |
| V 44 | | | | 0.35 |
| V 45 | 0.60 | | | |
| V 46 | 0.41 | | | |
| V 47 | 0.65 | | | |
| V 48 | 0.52 | | | |
| V 49 | 0.58 | | | |

**Table 1.** *Cont.*

| Variable | F 1 | F 2 | F 3 | F 4 |
|---|---|---|---|---|
| V 50 | 0.48 | | | |
| V 51 | 0.48 | | | |
| V 52 | 0.55 | | | |
| V 53 | 0.57 | | | |
| V 54 | 0.43 | | | |
| V 55 | 0.58 | | | |
| V 56 | 0.52 | | | |
| V 57 | 0.34 | | | |
| V 58 | 0.70 | | | |
| V 59 | 0.66 | | | |
| V 60 | 0.68 | | | |
| V 61 | 0.30 | | | |
| V 62 | 0.31 | | | |
| V 63 | 0.34 | | | |

Each of the factors extracted could be explained as a function of the different theoretical dimensions considered in the creation of the initial version of the instrument in the following manner:

- Factor 1 continued with the same denomination ("perception of the gender role of men"), corresponding to the dimension "perception of gender role" (men scale).
- Factor 2 continued with the same denomination ("fear of negative judgement"), corresponding to the dimension "Fear of negative judgement".
- Factor 3, "professional aspirations", corresponded to the dimensions "predisposition to leadership" and "relevance of the role of work".
- Factor 4 continued with the same denomination ("perception of the gender role of women"), corresponding to the dimension "perception of gender role" (women scale).

### 3.1.2. Study of Internal Consistency

To guarantee the reliability of the instrument, its internal consistency was analyzed with Cronbach's alpha coefficient, in general terms ($\alpha = 0.89$) as well as in the four factors extracted ($\alpha = 0.88$ in factor 1; $\alpha = 0.86$ in factor 2; $\alpha = 0.90$ in factor 3 and $\alpha = 0.91$ in factor 4). In every case, the results obtained made evident the high reliability of the questionnaire.

### 3.1.3. Content Analysis of the Items

During the application of the instrument, there were signs that some items were not clear for some of the students polled. As a result, the drafting of 11 items was modified with the objective of improving students' comprehension and to avoid negative connotations that could cause confusion in the response.

### 3.2. Study 2

### 3.2.1. Study of the Structure of the Instrument: Confirmatory Factorial Analysis (CFA)

In order to contrast and confirm the model extracted through the EFA, a CFA with maximum likelihood was conducted as a method of estimation. The results obtained are shown in Figure 1.

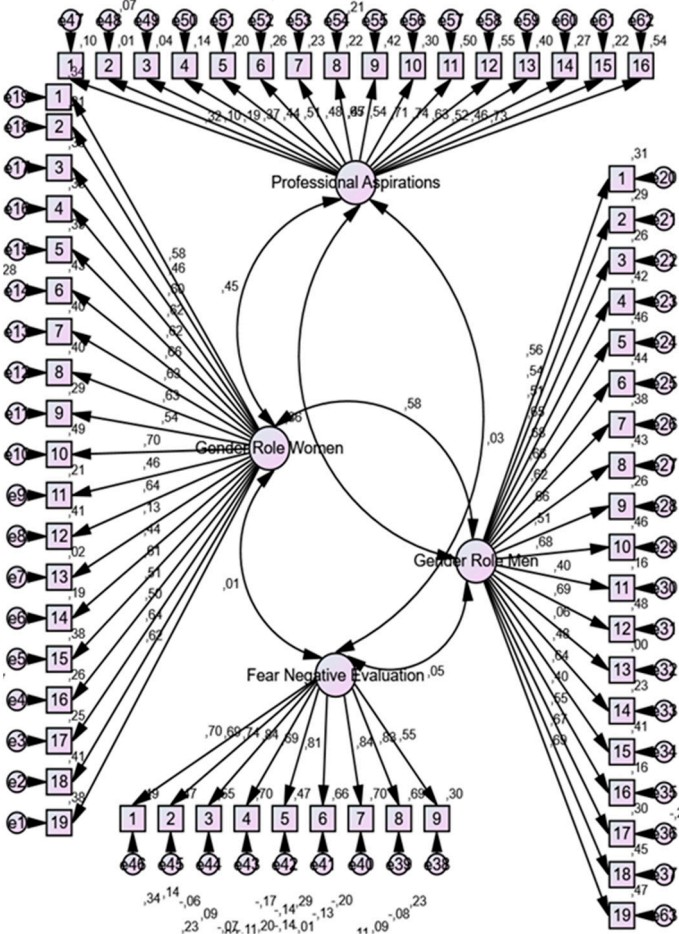

**Figure 1.** Four-factor model (CFA).

To evaluate the goodness of fit of the model identified, different indices were considered besides χ2 because of the latter's susceptibility to sample size (Herrero 2010): NFI (normed fit index), CFI (comparative fit index), IFI (incremental fit index), and RMSEA (root mean square error of approximation) (Freiberg Hoffmann et al. 2013; Byrne 2005).

The set of indices provided acceptable values (χ2 with a probability of 0.00, values of less than 0.06 in the case of RMSEA, and values near or higher than 0.95 in the case of NFI, CFI and IFI) (Byrne 2005; Arias 2008), which allowed confirming the model of factors proposed and guaranteeing the validity of the construct of the instrument (Table 2).

**Table 2.** Indices of fit of the model.

| Indices | χ2 | NFI | CFI | IFI | RMSEA |
|---------|-----|-----|-----|-----|-------|
| Value | 2472.86 (df. 1581) | 0.92 | 0.97 | 0.97 | 0.02 |

### 3.2.2. Study of Internal Consistency

In the application of the original version of the questionnaire (Lajo et al. 2008), the internal consistency of the instrument was analyzed according to dimensions (through Cronbach's alpha), with the results shown in Table 3. To guarantee the reliability of the instrument, the internal consistency was also analyzed in the present work, in general terms and by dimensions, with the results shown below:

**Table 3.** Internal consistency of the instrument.

| Dimension | QBP Original Version | QBP Current Version |
|---|---|---|
| Predisposition for leadership | $\alpha = 0.71$ ($n = 10$) | Unified dimension |
| Relevance of the role of work | $\alpha = 0.75$ ($n = 12$) | "Professional aspirations" |
| | | $\alpha = 0.84$ ($n = 16$) |
| Attitude towards social networks | $\alpha = 0.67$ ($n = 10$) | Dimension eliminated |
| Fear of negative judgement | $\alpha = 0.87$ ($n = 12$) | $\alpha = 0.92$ ($n = 9$) |
| Perception of gender role–women | Data not provided | $\alpha = 0.89$ ($n = 19$) |
| Perception of gender role–men | Data not provided | $\alpha = 0.89$ ($n = 19$) |
| Complete questionnaire | Data not provided | $\alpha = 0.91$ ($n = 63$) |

As shown in Table 3, in all the dimensions of the instrument in its current version, coefficients higher than 0.60 were reached, results that are considered to be acceptable (Thorndike 1997). This signified that as a whole, as well as in its different dimensions, the instrument had a high internal consistency. Compared to those from the original version of the questionnaire, the results from the newer version, for all the dimensions with data, were higher.

In order to corroborate this data, Cronbach's alpha was analyzed in the case of an element being eliminated, obtaining indices from $\alpha = 0.91$ with $\alpha = 0.91$. This showed that none of the items produced large variations in the general consistency of the questionnaire, and that all, without exception, had a high reliability.

### 3.2.3. Study on the Relationships among Variables

The relationships among the variables was studied with the difference in means analysis as a function of sex and the correlational analysis between dimensions.

The comparison of means showed that in the dimensions "fear of negative judgement" (U = 91,561.50, (202, 842), Z = 1.80, $p > 0.05$) and "gender role of men" (U = 84,477.50, (202, 842), Z = −0.17, $p > 0.05$), there were no significant differences as a function of sex. Men and women showed similar positions with respect to the fear of negative judgement and to how they perceived the role of men in the area of work.

However, the dimensions "professional aspirations" and "gender role of women" did show significant results. The dimension "professional aspirations" made evident that the women polled showed greater aspirations than the men in the area of work (U = 96,071.00, (202, 842), Z= −3.43, $p < 0.01$). More specifically, women had a mean of 4.64, while the mean for the men was 4.50. The dimension on the perception "gender role of women" had significant results (U = 98,589.00, (202, 842), Z = 4.14, $p < 0.01$), corroborating that women perceived the characteristics included in the dimension in themselves (with a mean of 4.62) more acutely than the men (who had a mean of 4.44).

The correlation analysis among dimensions, on its part, brought to light that there was a relationship among the dimensions, as already previewed by the CFA. Nevertheless, this relationship was not statistically significant in the dimension "fear of negative judgement". The results showed that the more acute relationship was found between the dimensions "gender role of women" and "gender role of men" (r = 0.33, $p < 0.01$). Other highlighted dimensions were found in the relationships between the dimension "professional aspirations" and "gender role of women" (r = 0.23, $p < 0.01$) and "gender role of men" (r = 0.18, $p < 0.01$).

## 4. Discussion

This article presents the process of adaptation and validation of the "Questionnaire of Barriers Perceived" (QBP) for a new research context, contributing, after the initial analysis of its psychometric characteristics, a double study (pilot and the definitive application of the instrument) that allowed for the examining of the instrument's internal consistency, the improving of its content, and the analysis of its structure in an exploratory and confirmatory manner.

The internal consistency indices obtained in both studies, for the scale in general as well as for each dimension, were high and superior to those found in the original version of the instrument (Lajo et al. 2008) and in other studies of similar nature (Mosteiro García and Porto Castro 2017). These results also indicated that besides being a reliable instrument, the QBP's reliability was improved with respect to the previous version of the questionnaire.

With respect to the evidence related to the instrument's validity, it should be pointed out that the content of the items was reviewed and modified as a result of its application in study 1. This process was complemented by two factorial analyses that guaranteed the validity of the questionnaire's construct. At the methodological level, this process was valuable, as it took into account the appropriate statistical procedures according to the sample size and distribution. However, it is still common to find some research studies in which this type of analysis is applied without the study showing previous assumptions related to the distribution and characteristics of the data collected, with the principal component analysis being the most reiterated (Bergman 2003; De Sola et al. 2003). The two sequential studies conducted also allowed for the creation of a hypothetical model starting with the data extracted from the EFA. This model was composed by four dimensions that explained 47% of the variance, a percentage that was higher than those in other studies (García Pérez et al. 2010) and that was then confirmed with other appropriate fit indices.

The dimensions that comprised the model confirmed were found to be interrelated and coherent with the theoretical structure considered in the initial creation of the instrument, although they were shaped and named, in part, in a different manner. These dimensions referred to the professional aspirations of the future teachers, with the fear of negative judgement, and with perceptions the future teachers had on the gender roles attributed to women and men. Other instruments reviewed referred to broader dimensions (sociocultural, relational, and personal), as in the cases of Baber and Tucker (2006) and García Pérez et al. (2010), or constructs associated with sexuality, religion, or values, in the case of De Sola et al. (2003), for example. In this case, however, more specific aspects were included that focused on the professional and academic sectors, just as in the research study by Bergman et al. (2014).

As for the relationships maintained among these dimensions, it was diverse, resulting in different degrees in each case. Thus, the professional aspirations of the students seemed to be related with the roles assigned to men and women, as shown by Cubillas Rodríguez et al. (2016) or Guil Bozal (2016), who identified that the roles attributed to women in their private lives or in the raising of children conditioned their professional aspirations and trajectories. The relationship established between the dimensions "gender role in men" and "gender role in women" was highlighted, corroborating how gender is a relational social construction, not a product of a person, and surges and develops through interaction (Rebollo-Catalán et al. 2011), in which not only the characteristics assigned to a gender, but those attributed by others, intervene. Lastly, the only dimension that was not found to be significantly related with the others was "fear of negative judgement". In this case, the expectations and preoccupations for the image that people in their surroundings have about themselves were not found to be associated with professional aspirations or perceptions of gender roles. This finding coincided with what has been shown in other studies, which have described the existence of a certain inverse relationship between failure and entrepreneurship (Asociación RED GEM España 2016).

In light of this discussion, it can thus be concluded that the questionnaire showed evidence of the validity of the content and the construct as well as an appropriate reliability. These findings indicate it to be a useful instrument for its application in similar contexts.

**Author Contributions:** Conceptualization, M.D.H.-A. and E.F.H.-P.; methodology, J.M.M.-G. and E.F.H.-P.; software, J.M.M.-G.; validation, J.M.M.-G.; formal analysis, J.M.M.-G. and E.F.H.-P.; investigation, M.D.H.-A., E.F.H.-P. and J.M.M.-G.; resources, M.D.H.-A.; data curation, M.D.H.-A.; writing—original draft preparation, M.D.H.-A.; writing—review and editing, J.M.M.-G.; visualization, E.F.H.-P.; supervision M.D.H.-A.; project administration, M.D.H.-A.; funding acquisition, not applicable. All authors have read and agreed to the published version of the manuscript.

**Funding:** This research was not external funding.

**Institutional Review Board Statement:** The study was conducted according to the guidelines of the Declaration of Helsinki.

**Informed Consent Statement:** Informed consent was obtained from all subjects involved in the study.

**Data Availability Statement:** Not applicable.

**Conflicts of Interest:** The authors declare no conflict of interest.

## Appendix A. Dimensions and Items of the Questionnaire

| Dimension | Items |
|---|---|
| **Professional aspirations** | 1. I hope to become a leader in my professional field<br>2. When I become established in my career, I would like to manage other employees<br>3. When I become established in my career, I would like to train other people<br>4. I hope to advance within the organization or business where I will work<br>5. I plan to continue with my education to achieve new objectives<br>6. I will try to become an expert in my field of work<br>7. I would like to be a key piece in the organization or business where I will work<br>8. Professional success is fundamental if one wants to be truly happy<br>9. I try to dedicate the time and energy necessary to make progress in my professional field<br>10. When it is about achieving my objectives, I am a very organized person<br>11. I plan to dedicate a significant part of my time to building my professional future and developing the necessary skills to make progress<br>12. Going the furthest possible in my professional life is, for me, an important motivation<br>13. I am the type of person who works very hard to achieve what interests me<br>14. In general, I plan my things, career, and work ahead of time to achieve my goals<br>15. The most important goal in my life is to have interesting and stimulating work<br>16. I will do everything possible so I can progress in my career |
| **Fear of negative judgement** | 1. I worry about what people think of me<br>2. I care about people having an unfavorable impression of me<br>3. I am afraid that people notice my defects<br>4. I worry about the impression I give to someone else<br>5. I worry that others do not agree with my way of acting<br>6. I worry that people find faults (or errors) in me<br>7. I care about the opinions others have about me<br>8. When I'm speaking with someone, I worry about what that person could be thinking about me<br>9. I worry about saying or doing the wrong things |
| **Perception of gender role (double dimension: women and men)** | 1. Ability to plan daily tasks<br>2. Emotional stability<br>3. Self-control mechanisms<br>4. Assuming work responsibilities<br>5. Interpersonal communication skills<br>6. Initiative and facing risks<br>7. Time dedicated to work<br>8. Satisfaction with work well done<br>9. Work and family reconciliation<br>10. Desire to succeed at work<br>11. In the workplace, they help promote people who are the same sex as they are<br>12. Collaboration with their peers<br>13. Accepting work with less pay very frequently<br>14. Valuing prestige and power within the organization<br>15. Worrying about the improvement of people in their surroundings<br>16. Care time for the family and the home<br>17. Personal self-esteem<br>18. Personal confidence on their own competencies<br>19. Expectations of work or professional success |

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
