# Peer review of "Psychometric Study of “Questionnaire of Barriers Perceived” (QBP) in Higher Education"

_socsci, doi:10.3390/socsci10120475_

Round 1

Reviewer 1 Report

Two areas for revision: First, there needs to be a clearer statement about why this instrument is needed. In what ways will use of this instrument to measure perceived gender barriers in ways that other measures cannot? In other words, is this instrument unique? Second, some minor editing is in order, especially the use of English language prepositions (i.e., substituting "to" when "with" is more accurate).

Author Response

First of all, we would like to thank the reviewer for the work carried out, which is essential for the publications to reach the appropriate level of quality for publication.

Focusing on the first of its questions, it should be noted, as mentioned in the theoretical framework of the article, that there are many instruments that measure factors related to gender barriers but none of them integrate the 5 dimensions discussed here, which are : the predisposition to leadership, the relevance of the work role, the fear of negative evaluation and the perception of the gender role in women and men. Furthermore, another added problem is that most of the instruments reviewed base their validation process on expert judgment; or, in an exploratory factor analysis (EFA), so the study presented here aims to provide future research on gender with a valid and reliable resource with high rates, which allows to provide rigor to the results obtained.

On the other hand, a revision of the writing of the article has been carried out, correcting the errors derived from the use of prepositions.

Reviewer 2 Report

The aim of the paper is present the results from a validation study of the “Questionnaire of Barriers Perceived” conducted among university students. It focuses on the factors and barriers that may influence the professional and academic choices and trajectories of men and women. These factors are related to gender stereotypes, professional aspirations, the fear of a negative judgement and the perceptions/awareness of gender roles. The results of the validation study reveal high goodness-of-fit index, validity and reliability of the scale. The extracted dimensions are coherent with the theoretical assumptions of the instrument and consistent with the findings from other empirical studies. My suggestion to the author/s is to add examples of statements associated with each extracted factor and to explain how the procedure of item content analysis has been applied and some statements modified.   

Author Response

First of all, we would like to thank the reviewer for the work carried out, which is essential for the publications to reach the appropriate level of quality for publication.

Focusing on your question, indicate that the procedure for analyzing the content of each of the items of the 5 factors is described in the results of study 1 and 2, respectively.

In the first case, the value of the determinant of the correlation of the matrix, the value of the KMO, the value of Barlett's test of sphericity, as well as the value of RMSR have been considered. Regarding the maintenance of the items, the value of each one of them in the communalities and in the matrix of rotated factors has been taken into account, eliminating those that presented weights below .30.

In the case of the confirmatory factor analysis, the fit indices of the χ2 NFI CFI IFI and RMSEA model were taken into account, as well as factor weights greater than .30

All the data is located in the article.

Reviewer 3 Report

The manuscript was a pleasure to read and offered important insights on the topic. However, there are a few limitations and shortcomings which undermine your arguments and influence the academic value of your article in a negative way:

  1. The theoretical framework seems to be underdeveloped, especially when considering the richness of the literature dedicated to the overarching topic. The literature review provides only a brief glimpse on the relevant existing scientific work, while it also lacks a consistent narrative which should have been used toward building a consistent argument. Furthermore, some of the claims made in the manuscript are not fully developed as valid arguments. A more comprehensive literature review should have been conducted, going into further details regarding the multiple types of gender discrimination in academia you have mentioned (both vertical and horizontal)
  2. Although the manuscript is written in a clear, concise and correct way, there are still small language mistakes (typos) or sections which could be rephrased in order to ensure better clarity.  I would recommend proofreading the manuscript one more time.
  3. The article presents one main methodological limitation regarding the sample on which the questionnaire was tested. 
    1. Both in study 1 and 2 the sample is mostly female (overrepresentation) and there seem to be no respondents from STEM fields (which is explainable considering the university where data was collected). However, the end result is the same as your sample does not seem to be representative for the target population.
    2. Ideally, the sample should be supplemented and more questionnaires filled in order to increase male representation and include more students with a background from STEM fields. However, if this is not possible at the moment, please try to address these issues more clearly and honestly in the methodological section.   
  4. More details should be offered in section 2.1. Instrument regarding how the instrument was actually designed and its connect6ion with previous literature. Which articles, books or instrument served as basis for its conception? Which are the theoretical underpinnings behind it?  
  5. The conclusions and policy recommendations developed based on the findings are, to some extent, general and cannot easily serve as true and convincing policy recommendations for decision makers. The manuscript fails to make good use of the results obtained and to translate them into original and specific policy recommendations that could help decision-makers develop sustainable public policies. More work should be done on the last section in this regard 
  6. The questionnaire should either be included in the manuscript as an Annex (ideally) or describer more thoroughly in the methodological section.   

Author Response

First of all, we would like to thank the reviewer for the work carried out, which is essential for the publications to reach the appropriate level of quality for publication.

Focusing on the first of its questions, we would have liked to further develop the theoretical framework of the article, however, as in most scientific journals, there are space limitations that force us to select certain information, giving more priority to the related with some of the instruments for measuring gender barriers, having to dispense with another, such as types of gender discrimination. However, it will be taken into account for the publication of the next article where we will focus on the results of descriptive, inferential and regressive analyzes.

Regarding the writing of the article, a new revision has been carried out, correcting some errors.

Regarding the sample, it should be noted that this research focuses on the education degrees of the University of XXX, having as its main characteristics their feminization, that is, the existence of a greater predominance of women over men. However, a paragraph has been added in the sample section where this aspect is addressed and is seen as a limitation.

Regarding the instrument, as specified in the corresponding section, it is an instrument prepared by Lajo et al. (2008), called "Questionnaire of perceived gender barriers" and that integrates 5 dimensions that are "Predisposition to leadership", "Relevance of the role of work", "Fear of negative evaluation" and "Perception of the role of gender in men and women ". Therefore, this instrument is not of its own creation, so it is invited to consult the original work to expand the theoretical foundations of its design, since, for reasons of space, it is impossible to address it in this article. However, it should be noted that one of the weaknesses of the original instrument lies in its validation process, since it was simply based on an expert judgment based on qualitative assessments, for which we believe it is convenient to carry out a more in-depth study, from a quantitative perspective. of research, which would provide it with suitable indexes of adjustment of the model that would allow future investigations to obtain much more rigorous results than with the initial version of the same, the main objective of this study.
Finally, the instrument is added to the document in the form of an annex, as requested.